# Anti-*Candida* Potential of Sclareol in Inhibiting Growth, Biofilm Formation, and Yeast–Hyphal Transition

**DOI:** 10.3390/jof9010098

**Published:** 2023-01-10

**Authors:** Chaerim Kim, Jae-Goo Kim, Ki-Young Kim

**Affiliations:** 1Department of Life Science, Gachon University, Seongnam 13120, Gyeonggi-do, Republic of Korea; 2Graduate School of Biotechnology, Kyung Hee University, Yingin 17104, Gyeonggi-do, Republic of Korea; 3College of Life Science, Kyung Hee University, Yongin 17104, Gyeonggi-do, Republic of Korea

**Keywords:** *Candida albicans*, sclareol, anti-fungal, apoptosis, virulence

## Abstract

Even though *Candida albicans* commonly colonizes on most mucosal surfaces including the vaginal and gastrointestinal tract, it can cause candidiasis as an opportunistic infectious fungus. The emergence of resistant *Candida* strains and the toxicity of anti-fungal agents have encouraged the development of new classes of potential anti-fungal agents. Sclareol, a labdane-type diterpene, showed anti-*Candida* activity with a minimum inhibitory concentration of 50 μg/mL in 24 h based on a microdilution anti-fungal susceptibility test. Cell membrane permeability with propidium iodide staining and mitochondrial membrane potential with JC-1 staining were increased in *C. albicans* by treatment of sclareol. Sclareol also suppressed the hyphal formation of *C. albicans* in both liquid and solid media, and reduced biofilm formation. Taken together, sclareol induces an apoptosis-like cell death against *Candida* spp. and suppressed biofilm and hyphal formation in *C. albicans*. Sclareol is of high interest as a novel anti-fungal agent and anti-virulence factor.

## 1. Introduction

*Candida albicans* is an important opportunistic fungal pathogen worldwide. *C. albicans* is normally commensal in the human gastrointestinal tract and skin. However, *C. albicans* can cause hospital-acquired infections from superficial infections of oral or vaginal mucous membranes to life-threatening systemic disease, especially in immunocompromised patients [1]. *Candida* infections in the United States are the fourth most common hospital-acquired bloodstream infections and a 40% mortality rate in patients with systemic candidiasis has been reported [2,3]. To solve these health problems, many studies have been conducted to treat *C. albicans* infection, but few drugs are currently available as front-line treatments. To make matters worse, some strains become more resistant against commercially used anti-fungal drugs such as azoles [4]. New drugs to treat *Candida* infection must be developed to overcome drug resistance [4,5,6].

Apoptosis is the most well-known form of programmed cell death, which occurs essentially for proper development of higher eukaryotes. In fungi, apoptosis occurs naturally during aging and reproduction, and is induced by environmental stresses and treatment with anti-fungal agents [7]. Apoptosis in fungi has some features including phosphatidylserine (PS) externalization, nuclear condensation, DNA fragmentation, reactive oxygen species (ROS) accumulation, mitochondrial membrane potential dissipation, and cell cycle arrest [8,9,10].

Biofilms are structured communities of fungi and have an important function for fungi to adhere to biotic and abiotic surfaces [11]. Biofilms also provide resistance to antimicrobial therapy and host immune defenses. If mature biofilms have already developed, it is difficult to completely remove them due to their resistance to medical methods using disinfectants or physical washing [12]. Therefore, inhibiting biofilm formation is a new and critical candidate to reduce fungal infection.

*C. albicans* has three biological phases including yeast, pseudohyphal, and hyphal forms. The transformation of *C. albicans* from yeast to hyphae has important roles in causing disease by escaping from the phagocytic activity of macrophages and increasing invasion activity into epithelial cells. Inhibition of hyphal formation is a good target for anti-fungal adjuvants to reduce fungal virulence [13].

Sclareol is a labdane diterpene derivative isolated from *Salvia sclarea* (Figure 1). Sclareol induces cell cycle arrest and apoptosis in human breast cancer cells and inhibits proliferation of osteosarcoma cells by apoptotic induction and loss of mitochondrial membrane potential [14,15,16,17]. Sclareol also can alleviate the severity of arthritis by modulating excessive inflammation [18].

In this study, sclareol was tested for new anti-*Candida* activity with apoptosis-like cell death and for reduction in virulence by inhibiting biofilm formation and morphological transition of *Candida* spp.

## 2. Materials and Methods

### 2.1. Fungi Strains and Growth Conditions

The following *Candida* strains were used in this study: *C. albicans* (KCTC7965), *C. tropicalis* var. *tropicalis* (KCTC17762), *C. parapsilosis* var. *parapsilosis* (KACC45480), *C. parapsilosis* (KACC49573), *C. glabrata* (KCTC7219), and *C. auris* (KCTC17809). They were purchased either from KCTC (Korean Collection for Type Cultures; Jeongeup, Republic of Korea) or KACC (Korean Agricultural Culture Collection; Suwon, Republic of Korea). All strains were stored as frozen stock with 20% glycerol at −70 °C. Strains were cultured on YPD containing yeast extract 10 g/L (BD Difco, Franklin Lakes, NJ, USA), peptone 20 g/L (BD Difco, Franklin Lakes, NJ, USA), 2% D-glucose (*w*/*v*) (Daejung, Gyeonggi-do, Suwon, Republic of Korea), and agar 15 g/L (Daejung, Gyeonggi-do, Suwon, Republic of Korea) at 30 °C for 24–48 h [14,19]. 

### 2.2. Anti-Fungal Susceptibility Microdilution Assay

A microdilution assay was performed based on the description of a previous work [20]. Sclareol (Sigma-Aldrich, St. Louis, MO, USA) was dissolved with dimethyl sulfoxide (DMSO, Junsei, Tokyo, Japan) as a 10 mg/mL stock solution. *Candida* strains were cultured in culture tubes overnight and diluted to absorbances at OD_600_ = 0.01. Amounts totaling to 100 µL of *C. albicans* cells were added into each well of a 96-well plate (SPL, Gyeonggi-do, Seoul, Republic of Korea) with serial two-fold dilution of sclareol (3.125–200 µg/mL). After incubating at 30 °C for 24 h, the minimum inhibitory concentration (MIC) of the compound was measured by the naked eye. After MIC detection, 50 µL of the cultures was spread onto a YPD agar plate and incubated for 24 h at 30 °C to obtain the minimum fungicidal concentration (MFC) value as a no-cell-growth determinant. Growth control without sclareol treatment and sterilized medium control were performed in each test [21].

### 2.3. Fungi Cell Growth Test

Culture conditions in this study were the same as those in the above-mentioned anti-fungal susceptibility microdilution assay. Sclareol (25 and 50 µg/mL) was added to 96-well plates (SPL Gyeonggi-do, Seoul, Republic of Korea) with *C. albicans*. Growth of *C. albicans* was measured by change in absorbance at OD_600_ = 0.05 using a microplate reader (BioTek Instruments, Winooski, VT, USA) at a time indicated in a previous work [22].

### 2.4. Membrane Permeabilization Assay

Propidium iodide (PI, Biotium, Hayward, California, SF, USA) staining was carried out as reported previously with some modifications [23]. Freshly grown *C. albicans* cells were diluted to 10^6^ CFU/mL in YPD medium and treated with indicated concentrations of sclareol (25, 50 and 100 µg/mL) at 30 °C for 2 h. Cells were washed twice with PBS and resuspended in 25 µg/mL of PI solution for 20 min at room temperature in the dark with shaking. After washing twice, the fluorescence was monitored and photographed with an EVOS FL Cell Imaging System (Thermo Fisher, Waltham, MA, USA) [24].

### 2.5. Measurement of Mitochondrial Membrane Potential

JC-1 (Koma Biotech, Seoul, Republic of Korea) staining was performed as reported previously with slight modifications to check mitochondrial membrane potential [24,25]. Freshly grown *C. albicans* cells were diluted to 10^6^ CFU/mL in YPD and incubated with 25, 50, and 100 µg/mL of sclareol at 30 °C for 2 h. Cells were washed with PBS twice and stained with 0.5 µM of JC-1 for 15 min at 37 °C. After washing twice, the fluorescence of JC-1 was monitored and photographed with an EVOS FL Cell Imaging System (Thermo Fisher, Waltham, MA, USA).

### 2.6. ROS Detection

Dihydroethidium (DHE) staining (Cayman chemical, Ann Arbor, MI, USA) was used to detect ROS generation in apoptotic cells [22]. *C. albicans* cells were diluted to 10^6^ CFU/mL in YPD, then incubated with indicated concentrations of sclareol or with 10 µM antimycin A (positive control) for 2 h. Cells were washed with a cell-based assay buffer. After discarding the assay buffer, a ROS staining buffer containing 0.1% DHE was added and incubated at 37 °C for 1 h. After removing the supernatant, a cell-based assay buffer was added to the cells. DHE-stained cells were measured by VICTOR2 (Perkin Elmer, Waltham, MA, USA) using an excitation wavelength of 480 nm and an emission wavelength of 580 nm.

### 2.7. Biofilm Formation Assay 

To assess biofilm formation, *C. albicans* cells were cultured on YPD medium overnight at 30 °C in a shaking incubator and washed twice with phosphate-buffered saline (PBS) [26,27,28]. Amounts totaling to 100 µL of *Candida* cell suspensions (OD_600_ = 0.5) in RPMI 1640 were dispensed in 96 microdilution wells with or without sclareol (3.125 to 100 μg/mL). The plates were incubated at 37 °C for 24 h to adhere, and after that, washed with PBS to remove nonadherent cells. Biofilms were stained with 100 μL of 1% aqueous crystal violet for 15 min and washed twice with PBS and destained with 200 μL of 33% acetic acid for 20 min. The 96-well plate measured the optical density of each well by using a microplate reader (BioTek Instruments, Winooski, VT, USA) at 570 nm. To calculate the relative biofilm formation rate of *C. albicans* according to different doses of compound, compound-free wells and biofilm-free wells were also included for control. The assay was conducted more than three times in triplicate.

### 2.8. Hyphal Transition Assay

To evaluate the effect of compound on the yeast-to-hyphae transition of *C. albicans*, YPD media with 10% fetal bovine serum (FBS, Gibco, NY, USA) was used [29]. *C. albicans* cells were cultured overnight in 3 mL YPD media at 30 °C. Overnight cultures were centrifuged at 3000× *g* for 10 min. After removing supernatant, cells were washed and prepared at an absorbance of OD_600_ = 0.05. Afterwards, cells were incubated with shaking (250 rpm) for 4 h at 37 °C with indicated concentrations of sclareol. YPD + FBS transition-inducing media was used as controls. Cultures were washed with distilled water and we checked hyphal formation using an ICX41 coupled with an OD400UHW-P digital microscope camera (Thermo, Waltham, MA, USA). The percentage of hyphae formed was calculated by the number of hyphae formed/the total number of *C. albicans* cells. 

Yeast-to-hyphae transition was also conducted using spider medium (10% nutrient broth, 10% D-mannitol, 2% K2HPO4, 20% agar) with indicated concentrations of sclareol [30,31,32]. Cells without compounds were used as control. The plates were incubated for 7 days at 37 °C and pictures were taken with an ICX41 coupled with an OD400UHW-P digital microscope camera [33,34,35]. 

### 2.9. Synergistic Anti-Fungal Activity Assay

To evaluate the anti-fungal synergistic activity, the fractional inhibitory concentration index (FICI) was used [36].

Simply, *C. albicans* cells were cultured overnight, resuspended with YPD medium at OD_600_ = 0.01, and added to each well of a 96-well plate. Miconazole and sclareol were added with serial two-fold dilution from a maximum of 100 µg/mL. Plates were incubated at 30 °C for 24 h. The synergistic effect of compound and miconazole was measured based on the FICI value [37]. The assay was conducted more than three times in triplicate.

FICI = (MICA in combination/MICA alone) + (MICB in combination/MICB alone). According to the following criteria, the effects of the anti-fungal drug combinations were classified: (1) FICI ≤ 0.5, synergistic effects; (2) 0.5 < FICI ≤ 1, additive effects; (3) 1 < FICI < 4, no interactions; (4) FICI ≥ 4.0, antagonistic effects. 

### 2.10. Quantitative Reverse Transcriptase PCR (qRT-PCR) Analysis

RNA isolation and quantitative RT-PCR were performed as described previously with some modifications [21,38]. Total RNA was extracted with TRIzol reagent (Life Technology, Thermo Fisher Scientific, Waltham, MA, USA) following the manufacturer’s instructions. cDNA was synthesized using 1 ug of RNA with reverse transcriptase (NanoHelix, Daejeon, Republic of Korea). qRT-PCR was performed with the 2X SybrGreen qPCR Mater Mix (CellSafe, Yongin, Republic of Korea). The expression level of tested genes was calculated using the formula 2-ΔΔCT. Primer sequences are listed in Table 1. *ACT1* was used for the internal control. Statistical qRT-PCR analyses were performed by using GraphPad Prism, version 3.00, for Windows (GraphPad Software, San Diego, CA, USA). 

### 2.11. Statistical Analysis

All data are expressed as the mean ± SD. Significant differences among the groups were determined using Student’s *t*-test and *p* < 0.05 was considered significant.

## 3. Results

### 3.1. Sclareol Inhibited the Growth of Candida spp.

The growth-inhibitory effect of sclareol against *Candida* spp. was evaluated. Sclareol showed an MIC value of 50 μg/mL for *C. albicans, C. auris*, and *C. parapsilosis* after 24 h (Table 2). To test the persistence of anti-fungal activity of sclareol to *Candida* spp., *Candida* was cultured for a further 72 h. The MIC of sclareol against *C. albicans* was 100 μg/mL after 48 h, and the value remained constant up to 72 h. The MFC value was 100 μg/mL for *C. albicans* after 24 h (Table 3). Treatment of 50 μg/mL of sclareol also inhibited *C. albicans* growth until 12 h, as indicated by a growth inhibition assay, but showed growth after 24 h (Figure 2). In summary, sclareol showed an anti-fungal effect against *Candida* spp.

### 3.2. Sclareol Reduced Membrane Permeability of C. albicans

The membrane permeability of *C. albicans* after treatment wiyj sclareol was detected using propidium iodide (PI) staining. After 2 h of sclareol treatment, the red fluorescence was increased depending on the concentration of sclareol (Figure 3a). In particular, the red fluorescence significantly increased by treatment of 100 μg/mL of sclareol. These results suggested that sclareol increased membrane permeability of *C. albicans*.

### 3.3. Sclareol Induced Apoptosis-like Cell Death through Disruption of Mitochondrial Membrane Potential and Increasing Intracellular Concentration of ROS

To further evaluate whether sclareol induced growth inhibition through apoptosis-like cell death, mitochondrial membrane potential and ROS generation were tested after sclareol treatment. Dissipation of mitochondrial membrane potential was detected by JC-1 staining. A JC-1 assay is a method of identifying mitochondrial membrane potentials using J-aggregate (JC-1) fluorescence. JC-1 dye is a lipophilic cation dye that permeates mitochondrial membrane well, having green fluorescence when present as a monomer and red fluorescence when present in the J-aggregates. In other words, when apoptosis occurs, the JC-1 cannot pass through the membrane and becomes green, and if it is alive, it penetrates the membrane well and becomes red [25]. *C. albicans* treated with sclareol showed a significant loss of red fluorescence compared with the untreated control (Figure 3b). This result indicates that sclareol inhibited the growth of *C. albicans* by abnormally altering the mitochondrial membrane potential. To support this result, ROS production was checked by DHE staining. Treatment with sclareol for 2 h increased the levels of intracellular ROS in a dose-dependent manner (Figure 3c). ROS generation in *C. albicans* started to remarkably increase at exposure to 100 µg/mL of sclareol [40,41,42,43,44,45,46].

To determine the molecular mechanism of sclareol in *C. albicans*, mRNA expression of *Candida* apoptosis-related genes was tested (Table 1). *MCA1* is a cysteine protease involved in apoptosis in response to stresses and a homologue of a mammalian caspase in *C. albicans*. *HSP90* is a molecular chaperone in the stress response of *C. albicans* and orchestrates echinocandin resistance via regulating the calcineurin pathway, which is involved in apoptosis [47,48,49,50]. *YPK1* is a serine/threonine protein kinase that affects a variety of cellular activities, including sphingolipid homeostasis [51,52,53]. The expression of three key genes, *MCA1, HSP90*, and *YPK1*, was dramatically increased by sclareol treatment in a dose-dependent manner (Figure 3d). The *SOD* gene family provides instructions for creating an enzyme called superoxide dismutase, which is related to ROS. The expression of *SOD2* and *SOD5* genes was increased by sclareol treatment in a dose-dependent manner, excluding the *SOD1* gene, whose expression was dose-dependently reduced (Figure 3e) [54]. Based on the above results, it is suggested that sclareol could induce apoptotic-like cell death against *C. albicans* (Figure 3f).

### 3.4. Sclareol Inhibited the Biofilm Formation of C. albicans

The inhibition of biofilm formation of *C. albicans* by sclareol was evaluated to determine whether sclareol also reduces the virulence of *Candida* or not. Sclareol dose-dependently inhibited the biofilm formation of *C. albicans* (Figure 4a). The expression of biofilm-related factors including *ZAP1, ADH5, CSH1, TPO4*, and *CAN2* was significantly decreased after treatment with 6.25 to 50 µg/mL of sclareol, which supports the previous result (Figure 4b, Table 1). *ZAP1* is a negative regulator of a major matrix component, soluble β-1,3 glucan [55,56,57]. *ADH5* represents extracellular matrix genes and promotes matrix production. The expression of *CSH1* is related to inhibition of structured communities of fungi. *TPO4* plays a role in biofilm formation and *CAN2* may participate in nutrient signaling pathways important for biofilm formation [58,59,60,61].

### 3.5. Sclareol Inhibited the Hyphal Formation of C. albicans

The inhibition of hyphal formation of *C. albicans* by sclareol was evaluated because biofilm formation is related with the morphological transition of *C. albicans*. Sclareol dose-dependently inhibited the hyphal formation of *C. albicans* (Figure 5a,c). Hyphal formation of *C. albicans* in spider medium is also inhibited by treatment of 50 μg/mL of sclareol (Figure 5b).

To support the result, the expression of *Candida* hyphae-related genes was tested (Table 1). After treatment with 6.25 to 50 µg/mL of sclareol, the expression of hyphae-associated factors in *C. albicans* was significantly decreased (Figure 5d). The expression of *ALS3* mediates adherence to epithelial cells by encoding a cell surface protein. The expression of *CYR1* is associated with the Ras1–cAMP–PKA pathway. *ECE1* expression is correlated with the extent of cell elongation and *ECE1* plays a role in the process of hyphal formation. The expression of *TPK1* needs to control hyphal formation [62,63,64].

### 3.6. Sclareol Synergistically Inhibited the Growth of C. albicans with Miconazole

The FICI value between sclareol and miconazole against *C. albicans* was 0.31 (Table 4). Sclareol with miconazole synergistically inhibited the growth of *C. albicans* compared with the miconazole or sclareol alone group. Even though the MIC value of sclareol was 50 μg/mL, 3.125 μg/mL of sclareol combined with 0.78 μg/mL of miconazole showed significant inhibition of *C. albicans* growth. The FICI value was determined from three independent experiments [65,66].

## 4. Discussion

The low efficacy, high toxicity, and drug resistance of currently available anti-fungal agents necessitate the search for next-generation anti-fungal agents [40]. Many natural products have been reported to exhibit anti-fungal activity by inducing apoptosis [41,42,43]. Apoptosis is a form of programmed cell death and important for homeostasis and maintenance by eliminating mutated, damaged, infected, or superfluous cells without an inflammatory reaction [44]. In addition to metazoans, apoptosis has also been found in unicellular organisms. The production of ROS is one of earliest changes associated with apoptosis and ROS are necessary and sufficient to induce apoptosis in yeast [45,46]. Previous studies have shown that sclareol and its derivatives have remarkable anti-fungal activities against *Botrytis cinerea* [47]. However, the potential mechanisms explaining the anti-fungal action have not yet been fully explained.

Sclareol showed inhibition of *C. albicans* growth through increasing cell death. It was found that the number of dead cells (using PI staining) was increased and mitochondrial membrane potential was broken (using JC-1 staining) by treatment of sclareol. Mitochondria are both the origin and target of ROS and play an essential role in apoptotic processes [49]. These results indicated that sclareol caused mitochondrial dysfunction and ROS accumulation and could induce apoptosis in *C. albicans*.

Sclareol also exhibited synergistic anti-fungal activity with miconazole, which is a well-known imidazole anti-fungal agent. Miconazole inhibits ergosterol synthesis which influences the membrane permeability of *Candida*. This might enable sclareol to invade *Candida* more and result in synergistic activity [50]. The other possible mechanism lies in the notion that miconazole can also increase ROS generation which could synergistically affect *Candida* growth [51]. However, the detailed mechanism of synergistic anti-fungal activity must be examined in the future with other commercially available anti-fungal agents and natural compounds.

Apoptosis is a complex process involving multiple factors, including *MCA1*, *HSP90*, and *YPK1* [52]. *MCA1* is a central regulator of apoptosis and is involved in releasing ROS in fungi [9]. *MCA1* encodes a human homologue of caspase. The calcineurin pathway was required for H_2_O_2_-induced *C. albicans* apoptosis. *HSP90* was activated and CaMCA1 expression was elevated; this resulted in increased caspase activity and, thus, induced apoptosis [52]. Sphingolipids (SL) are a group of lipids presented in eukaryotic plasma membrane and involved in pivotal cellular processes such as apoptosis in fungi [53]. SL biosynthesis is regulated by paralog *YPK1* protein kinases, a member of the AGC kinase subfamily [54]. The *SOD* gene codes superoxide dismutases (SODs), which are in the cytoplasm, chloroplasts, and mitochondria of eukaryotic and prokaryotic cells. SODs are critical antioxidant enzymes; they protect organisms from ROS caused by adverse conditions. The expression of *MCA1, HSP90*, *YPK1*, and most SOD genes (except *SOD1*) were increased by treatment of sclareol, and this result supports the notion that sclareol induces apoptosis via increasing and accumulating ROS in *C. albicans* [56].

The biofilm of fungi resists anti-fungal therapies and host defenses, and so it is necessary to suppress the production of a microbial-produced extracellular matrix. Biofilm formation is a complex process involving multiple factors, including *ZAP1*, *ADH5, CSH1, TPO4*, and *CAN2. ZAP1* is the zinc response transcription factor and regulates the expression of genes for matrix formation in *C. albicans* [58,60]. *ADH5* supports a matrix-stimulatory signal and may provide hexose for β-1,3 glucan synthesis. *CSH1* represents *ZAP1*-activated genes, and predicted alcohol dehydrogenases *TPO4* encode a putative transporter similar to polyamine and major facilitator superfamilies of drug transporters. Polyamines are essential for normal cell growth, and polyamine levels are carefully regulated in higher eukaryotes. *CAN2* may be correlated with drug and toxic substance transport or nutrient sensing and signaling pathways. Both *TPO4* and *CAN2* are involved in transporting small molecules which affect biofilm formation in *C. albicans* [60]. As a result, the formation of biofilms should be suppressed by treatment with sclareol.

*C. albicans* has three biological phases in yeast, pseudohyphal, and hyphal forms. The morphological transition of *C. albicans* from yeast to hyphae has an important role in causing disease by escaping from the phagocytosis of macrophages and invading epithelial cells. Hyphal formation is a complex process involving multiple factors, including *ALS3*, *CYR1, ECE1*, and *TPK1. ALS3* mediates adherence to epithelial cells by encoding a cell surface protein. In *C. albicans*, disruption of both copies of *ALS3* reduces adherence to endothelial cells and overexpression of this gene increases adherence. *ALS3* is also required for adherence of the hyphal form of *C. albicans* [62]. Yeast-to-hyphae transition in *C. albicans* requires a rapid activation of the adenylyl cyclase *CYR1* to generate a cAMP spike. *CYR1* acts as sensors of hyphae-inducing molecules. *ECE1* is a key gene for hyphal development and its expression has been shown to be correlated with cell elongation and biofilm formation. *TPK1* is essential for the derepression of branched-chain amino acid biosynthesis genes that have a role in the maintenance of iron levels and DNA stability within mitochondria. *TPK1* allows hyphal formation on solid inducing media. The four key genes in hyphal formation, *ALS3, CYR1, ECE1*, and *TPK1*, were downregulated by treatment with sclareol. Sclareol inhibits *C. albicans* hyphal formation by suppressing gene expression.

In summary, sclareol inhibits the growth of *C. albicans* by inducing apoptosis-like cell death and reduces the virulence by inhibiting biofilm formation and hyphal transition [67]. This, sclareol is useful for new anti-*Candida* adjuvants and can act as a scaffold to develop new anti-fungal molecules (Figure 6).

## 5. Conclusions

Sclareol inhibits the growth of *C. albicans* by inducing apoptosis-like cell death via induction of production of ROS, disruption of mitochondrial integrity, and induction of expression of apoptosis-related genes. Sclareol also reduces the virulence by inhibiting biofilm formation and hyphal transition. The anti-fungal efficacy of sclareol may lead to the development of novel anti-fungal agents with low virulence of *C. albicans*.

## Figures and Tables

**Figure 1 jof-09-00098-f001:**
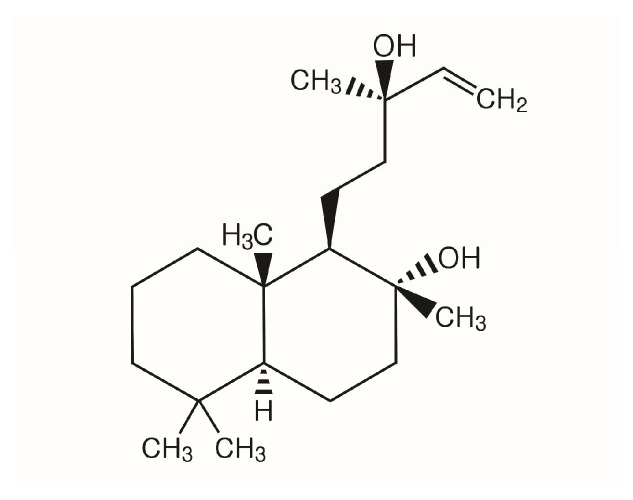
Chemical structure of sclareol.

**Figure 2 jof-09-00098-f002:**
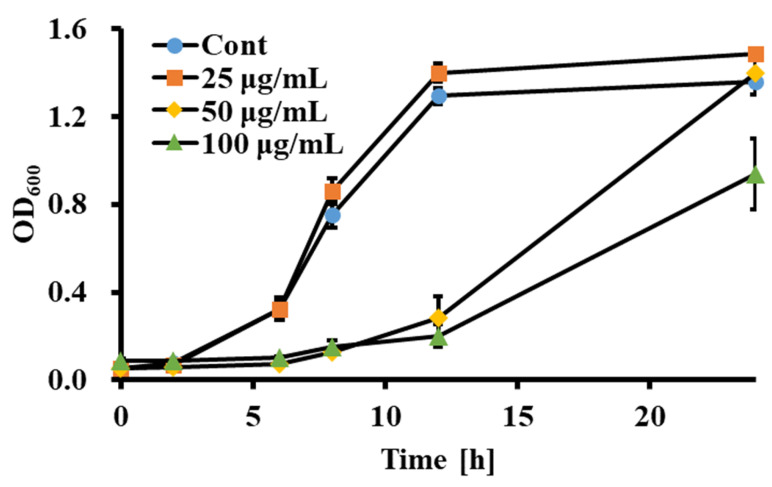
The growth of *C. albicans* was inhibited by treatment with sclareol. *C. albicans* (OD_600_ = 0.05) cells were cultured with 25–100 μg/mL of sclareol for 24 h at 30 °C. The growth was monitored by change in absorbance at 600. Data are presented as mean ± SD from three independent experiments.

**Figure 3 jof-09-00098-f003:**
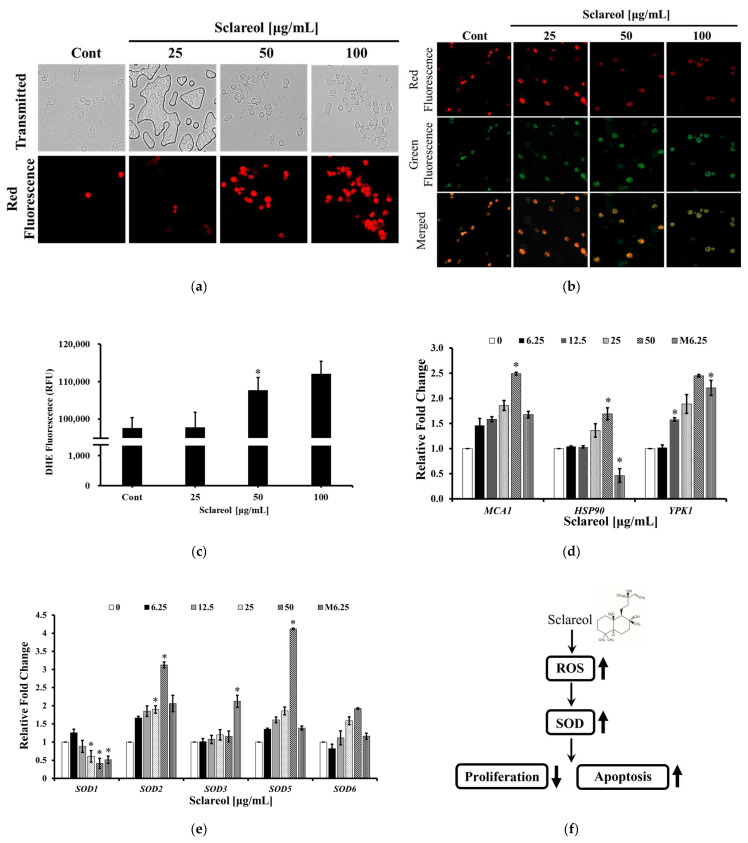
Sclareol-induced apoptosis-like cell death of *C. albicans*: (**a**) Propidium iodide (PI) staining showed fungal apoptosis-like cell death, a response caused by sclareol. Before fluorescent staining, *C. albicans* was cultured in YPD, and then treated with indicated concentration of sclareol for 2 h at 30 °C. (**b**) Sclareol induced depolarization of mitochondrial membrane potential in *C. albicans*. Visualization by fluorescent microscopy of *C. albicans* stained with JC-1. *C. albicans* was cultured in YPD, and then treated with indicated concentrations of sclareol for 2 h at 30 °C. Cells were washed with PBS and stained with 0.5 μM of JC-1 for 15 min. (**c**) Sclareol-induced apoptosis in *C. albicans* by increasing ROS levels. The graph shows intracellular ROS levels in 25–100 μg/mL sclareol-treated cells at 2 h. Cells were stained with dihydroethidium and analyzed by reading excitation and emission wavelengths of 480 and 580 nm, respectively. Data are presented as the mean ± SD of triplicate experiments. * *p* < 0.05 was considered significant. (**d**) Sclareol induced the expression of apoptosis-related genes of *C. albicans*. mRNA of *C. albicans* (KCTC7965) was obtained after 24 h treatment. A total 6.25–50 μg/mL of sclareol was used for treatment. M6.25 represents 6.25 μg/mL of miconazole. Data are presented as the mean ± SD of triplicate experiments. * = *p* < 0.05 was considered significant. (**e**) Sclareol induced the expression of *SOD* genes of *C. albicans*. Data are presented as the mean ± SD of triplicate experiments. * *p* < 0.05 was considered significant. (**f**) The predicted pathway of apoptosis in *C. albicans* induced by sclareol.

**Figure 4 jof-09-00098-f004:**
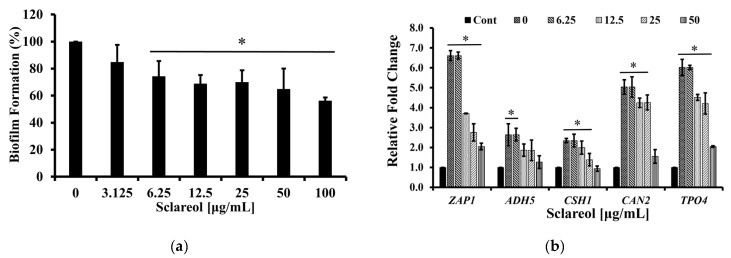
Sclareol inhibited the biofilm formation of *C. albicans*. (**a**) *C. albicans* biofilm was formed in medium with or without sclareol (3.125–50 μg/mL) at 37 °C for 24 h. Data are presented as the mean ± SD of triplicate experiments. * *p* < 0.05 was considered significant. (**b**) Sclareol inhibited the expression of genes related with biofilm formation in *C. albicans*. mRNA of *C. albicans* (KCTC7965) was obtained after 24 h treatment in biofilm formation condition. Data are presented as mean ± SD from three independent experiments. * *p* < 0.05.

**Figure 5 jof-09-00098-f005:**
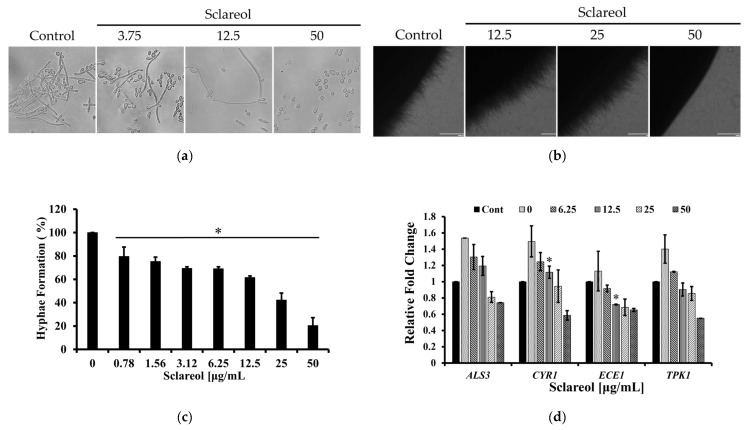
Sclareol inhibited the hyphal formation of *C. albicans*. (**a**) Sclareol inhibited the hyphal formation of *C. albicans* in broth media. *C. albicans* hyphae was formed in a medium supplemented with FBS with or without sclareol (3.75–50 μg/mL) at 37 °C for 24 h. (**b**) Sclareol inhibited the filamentation of *C. albicans* on spider medium. *C. albicans* cells were cultured on the spider media plates with sclareol for 7 days at 37 °C. (**c**) Percentage of hyphal form of *C. albicans* in broth media. Data are presented as the mean ± SD of triplicate experiments. * *p* < 0.05 was considered significant. (**d**) Sclareol inhibited the expression of genes related with hyphal formation in *C. albicans*. Data are presented as mean values ± SD of triplicate experiments. * *p* < 0.05 was considered significant.

**Figure 6 jof-09-00098-f006:**
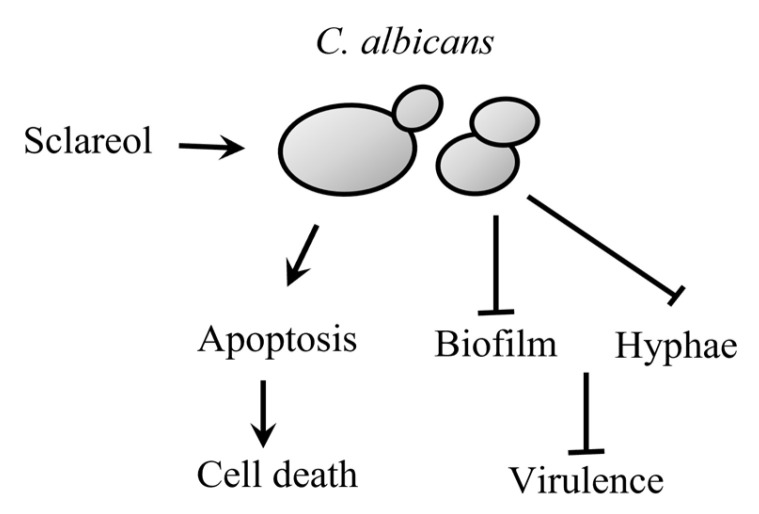
The schematic diagram of anti-fungal effect of sclareol against *C. albicans*.

**Table 1 jof-09-00098-t001:** List of qRT-PCR primers.

Gene	Primer Sequence	Function	References
*ACT1*	F: TAGGTTTGGAAGCTGCTGG R: CCTGGGAACATGGTAGTAC	Control	[39]
*MCA1*	F: TATAATAGACCTTCTGGAC R: TTGGTGGACGAGAATAATG	Apoptosis	In this study
*HSP90*	F: GGGAATCTAACGCTGGTGGTAA R: TTCGGTTTCTGGAACTTCTTTT	Apoptosis	In this study
*YPK1*	F: CAACACAACACAGTAGCACC R: GTTGTGGATAAAGGTGGTTCG	Apoptosis	In this study
*SOD1*	F: TTAAAGCTGTCGCTGTTGTC R: AATATGGAAACCTCTCAAGGC	Antioxidant	In this study
*SOD2*	F: AACTTGGCTCCTGTCTC R: TATCACCATTGGCTTTG	Antioxidant	In this study
*SOD3*	F: TATCACCATTGGCTTTG R: TATCACCATTGGCTTTG	Antioxidant	In this study
*SOD5*	F: ACGAGGGACACGGCAATGCT R: ACGAGGGACACGGCAATGCT	Antioxidant	In this study
*SOD6*	F: GACCCCGACCCACCTCAACAA R: GGGTAGCAAGGAGTGCCGGT	Antioxidant	In this study
*ZAP1*	F: ATCTGTCCAGTGTTGTTTGTA R: AGGTCTCTTTGAAAGTTGTG	Biofilm	[39]
*ADH5*	F: ACCTGCAAGGGCTCATTCTG R: CGGCTCTCAACTTCTCCATA	Biofilm	[39]
*CSH1*	F: CGTGAGGACGAGAGAGAAT R: CGAATGGACGACACAAAACA	Biofilm	[39]
*TPO4*	F: GCTGCTACCAATGTCAGTCC R: ACGGAGCTATCCGAATCGTC	Biofilm	In this study
*CAN2*	F: GCGGAATGGATATGCATGGG R: CGGATTGCTCTTGGAGAAGC	Biofilm	[39]
*ALS3*	F: GGTTATCGTCCATTTGTTG R: TTCTGTATCCAGTCCATCT	Hyphae	[39]
*CYR1*	F: GTTTCCCCCACCACTCA R: TTGCGGTAATGACACAACAG	Hyphae	[39]
*ECE1*	F: ACAGTTTCCAGGACGCCAT R: ATTGTTGCTCGTGTTGCCA	Hyphae	[39]
*TPK1*	F: CCAACGATTCCCTACTCCAG R: GCAATATAATCTGGAGTCCCAC	Hyphae	In this study

**Table 2 jof-09-00098-t002:** Anti-fungal effects of Sclareol against *Candida* spp.

*Candida* spp.	Sclareol [μg/mL]		
	24 h	48 h	72 h
*C. albicans* (KCTC7965)	50	100	100
*C. auris* (KCTC17809)	50	>100	>100
*C. glabrata* (KCTC7219)	>100	>100	>100
*C. parapsilosis* var. parapsilosis (KACC45480)	50	>100	>100
*C. parapsilosis* (KACC49573)	>100	>100	>100
*C. tropicalis* var. tropicalis (KCTC17762)	>100	>100	>100

**Table 3 jof-09-00098-t003:** Sclareol showed fungistatic activity against *C. albicans*.

*Candida* spp.	Sclareol [μg/mL]	
	MIC	MFC
*C. albicans* (KCTC7965)	50	100

**Table 4 jof-09-00098-t004:** Sclareol synergistically inhibited the growth of *C. albicans* with miconazole.

*Candida* spp.	MIC(μg/mL) of Miconazole		MIC(μg/mL) of Sclareol		FICI	Synergy
	Alone	With Sclareol	Alone	With Miconazole		
*C. albicans* (KCTC7965)	3.125	0.78	50	3.125	0.31	Synergy

## Data Availability

Not applicable.

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
