# Peer review of "Anti-Candida Potential of Sclareol in Inhibiting Growth, Biofilm Formation, and Yeast–Hyphal Transition"

_jof, 2023, doi:10.3390/jof9010098_

Round 1
Reviewer 1 Report
In this work authors analyse the anti-Candida activity of the drug sclareol. First, in a microdilution assay, the MIC and the MFC of the drug were estimated in 50 and 100 µg/ml respectively. However, MIC value significantly decreased when C. albicans was incubated at the same time with sclareol and miconazole. So, the two compounds exert a synergistic effect on each other. In this case, experiments have been carried out according to standar protocols. However, some details are missing in the description of the methodology. For instance, the number of replicates in the microdilution assay. In any case, MIC and MFC values are reliable.
However, some results and conclusions should be reconsidered by the authors in order to increase the scientific quality of the work:
1) About the effect of sclareol on apoptosis (fig. 3), the PI staining (fig. 3a) indicate that cell permeability is increased in presence of the drug. However, the authors just conclude the opposite (section 3.2). Although some signs of apoptosis are evident (intracellular levels of ROS, fig. 3c), the mitochondrial membrane depolarization should be quantified by FACS (fig. 3b). On the other hand, the sentence on lines 228-230 about the effect of the drug on SOD genes expression is incorrect. Only the expression of SOD2 and SOD6 are significantly altered by sclareol.
2) One of the most notable conclusions of the researchers is that sclareol inhibits the formation of biofilms. Again, some details of the methodology are missing (number of replicates in the 96-well plates). In addition, a statistical analysis of the data in fig. 4a would be necessary.
3) With regard to the effect of the drug in the yeast-hypha transition, photos of growth in liquid medium supplemented with FBS don´t have enough resolution (fig. 5a).
Finally, it would be interesting that the authors discuss about a possible relationship between the different effects of the sclareol in C. albicans. Might be the biofilm formation inhibition a consequence of the effect on growth?
Author Response
"Please see the attachment."

Reviewer 2 Report
In the present work, authors propose using sclareol as treatment for Candida spp. infections, especially C. albicans. Some points need to be adressed in order to this manuscript be suitable for publication.
Firstly, English must be improved. Several corrections may be done, such as:
- Page 1 line 32: remove “fluconazole” and “The”
- Page 1 line 35 (and along the manuscript) : change “fungal” to “fungi”
- Page 1 line 42: provide
- Page 1 line 44: remove “of”
- Page 2 line 48: phagocytic
- Page 2 line 74: were
- Page 3 line 108: cells were washed
Other points:
- Page 1 line 44: explain “medical methods”
- Page 3 line 119: insert incubation time
- Insert information about propidium iodide supplier
- Insert information about microscope supplier
- Authors said that sclareol inhibited the growth of C. albicans after 24h, with a MIC value of 50 μg/ml. However, figure 2 shows fungal growth of about 50%. Authors must explain this discrepancy.
- Figures 3c, 4a, and 5c lack statistical analysis.
- Authors must explain how the fluorescent probes work (if they enter intact or damaged cells; the difference between green and red JC-1; where propidium iodide binds, etc).
- Authors must improve the quality of figures 5a and 5b.
- Check italics along the text.
- Authors must discuss the results of synergy experiments. Why only miconazole was tested? How sclareol could be improving miconazole activity?
- At synergy experiments, MIC alone and MIC combined were defined as 100% growth inhibition?
- Have the toxicity profile of sclareol been reported in literature?
- Sclareol showed antifungal activity at 50 – 100 μg/ml. Nonetheless, fungal growth, biofilm formation and yeast-to-hyphae transition were not completely inhibited. Authors must explain why they consider that sclareol could be a lead compound to treat Candida infections. In my opinion, the results do not corroborate it. It would be better to say that sclareol could be used as a scaffold to develop new antifungal molecules.
- The experiments lack controls, such as H2O2 at ROS assay and sodium azide (or other compound) at JC-1 assay.
Author Response
"Please see the attachment."

Round 2
Reviewer 1 Report
In this new version of the paper, the authors have satisfactorily addressed the considerations made by both reviewers. However, the following minor points should be taken into account before publication.
Lines 190-195: the paragraph in red to explain the result of the JC-1 assay is in an incorrect position in the text. As the authors indicate, this is an assay to measure mitochondrial membrane potential and should therefore be included in section 3.3.
Line 238: although the authors have corrected the previous version, they still claim that there is an increase in SOD3 expression in the presence of sclareol. However, their statistical analysis indicates that this increase is not significant (the asterisk is only present in the treatment with miconazole). I considere that this result should be explained more precisely.
Author Response
"Please see the attachment."

Reviewer 2 Report
Authors made the changes I requested. Thus, in my opinion, the manuscript is suitable for publication.
Author Response
Thank you!